# Social contexts and black families' engagement in early childhood programs

Jane Leer [1,2*], Imari Z. Smith[1], Zoelene Hill[3], Lisa A. Gennetian[1]

**1** Sanford School of Public Policy, Duke University, Durham, North Carolina, United States of America,
**2** Department of Psychology, San Diego State University, San Diego, California, United States of America,
**3** New York Academy of Medicine, New York, New York, United States of America

* jleer@sdsu.edu

## Abstract

In the U.S., the federal government and dozens of cities have invested in home visiting programs intended to be universally available at scale to support caregivers of young children. Evaluations find that participation in these programs reduces maternal mortality, improves maternal mental health, and supports children's healthy development. Yet, many parents of young children who are invited to participate in home visiting programs do not enroll. This study fills gaps in the literature by examining how the broader social context affects Black families' engagement in home visiting programs. Via focus groups, survey data from a socioeconomically diverse sample of Black parents across the U.S., and a pre-registered field experiment, we capture views of and experiences with early childhood home visiting programs. We assess the responsiveness of these views to the broader social context and examine implications for interest and participation in home visiting programs. Focus group participants described benefits of home visiting while also expressing concerns about being unfairly judged about their parenting practices and the risk of a home visit resulting in child welfare system involvement. One out of four Black parents surveyed associated the term "home visit" with surveillance (i.e., government scrutiny of parenting), and associating "home visit" with surveillance was empirically correlated with lower participation in home visiting programs. Further, our pre-registered survey experiment showed a causal link between surveillance fears and home visit engagement. Reading a news article about a family's experience with the child welfare system decreased interest in home visiting among Black parents, while labeling a program as "new baby wellness" rather than "home visit" increased interest. Collectively, the findings point to ways in which the broader social context of parenting/parental surveillance negatively affects Black parents' participation in early childhood home visiting programs despite their interest.

## Introduction

The period immediately after birth, sometimes referred to as the "fourth trimester," is pivotal for connecting families to parental and infant health support [1]. To address this sensitive period of development, the U.S. government allocates $500 million annually towards home

**Data availability statement:** Quantitative data and analysis syntax for studies 2 and 3 are publicly available from the Open Science Framework: https://osf.io/tv2s6/. Citation: Leer, Jane, Lisa A Gennetian, Imari Smith, and Zoelene Hill. 2024. "Spill-over Effects of Systemic Racial Biases in the Child Welfare System on Black Parents' Engagement in Parenting Support Services Untitled." OSF. June 6. doi: 10.17605/OSF.IO/TV2S6. Qualitative data from study 1 contains identifiable participant information (e.g., voice and video recordings, transcripts that contain identifiable information such as location of residence and hospital used for labor and delivery) and are not available publicly per the Duke University Institutional Review Board (protocol 2022-01115). Data requests for study 1 can be sent to the Center for Child and Family Policy at the Sanford School of Public Policy, Duke University (contact: Dr. Jennifer Lansford, lansford@duke.edu).

**Funding:** This research was supported by an internal award from Duke University (2020 seed grant from All Children and Babies Thrive at Duke University) to Lisa A. Gennetian and a 2022 State and Local Policy Work Grant from the Society for the Psychological Study of Social Issues to Jane Leer. The funders did not play any role in study design, data collection and analysis, decision to publish, or preparation of the manuscript.

**Competing interests:** The authors have declared that no competing interests exist.

visiting programs targeting the caregivers of newborns and young children [2]. High-quality home visiting can reduce maternal and child mortality [3], improve maternal mental health [4–7], and strengthen parent-child interactions [4,5,8,9].

By connecting parents to vital clinical, social, and support resources, home visiting programs have the potential to support Black families with young children and reduce racial disparities in maternal and infant health [3,10,11]. However, program uptake poses a barrier, hindering the intended population-level impact [12]. In 2023, only 25% of families served by federally funded home visiting programs were Black, despite Black families' more elevated representation among populations deemed high priority for home visiting (e.g., those living in poverty, single parents, and parents under 21) [2].

Efforts to increase home visiting participation have deployed implementation practices grounded in cultural competence, which have been shown to enhance parents' persistence in programs once enrolled [13–15]. Less attention has been paid to *enrollment* in home visiting, however, despite many families either refusing services outright or agreeing to participate but then not being available to schedule a visit [16]. To date, research on enrollment decisions has emphasized individual- and micro-level factors such as low levels of parental educational attainment [17], fear of judgment for help seeking [18], housing instability [19], and perceived lack of need [12,19].

This study shifts attention to the potential role of the broader policy climate within which home visits are delivered. In the U.S., the policy climate toward Black families is defined by racialized surveillance of family life via the child welfare system, which disproportionately investigates and more severely adjudicates cases with Black families [20–22]. One in three children overall experience an investigation by Child Protective Services (CPS), compared with nearly *half* of Black children [23]. Investigations often occur in response to reports of suspected child abuse or neglect filed by mandated reporters such as teachers and social service providers. CPS then initiates a "home visit," in which agency representatives visit the home to identify potential abuse or neglect. Racial bias has been shown to be present at all steps—from referrals to CPS by mandated reporters to subsequent judgments of parenting following a CPS home visit [22,24–27]. Even when reports are deemed unsubstantiated, experiencing a CPS investigation has long-lasting negative effects on child and family well-being [28–30].

Thus, deciding *not* to participate in home visiting can be viewed as an adaptive response to an adversarial climate toward Black families. Experiences with direct racism or with vicarious racism, which occurs indirectly by hearing about or seeing racist acts committed against members of one's racial group, can lead to hypervigilant behaviors such as avoiding situations where discrimination may occur [31–33]. Qualitative studies document how concerns about CPS surveillance shape the types of information mothers share with healthcare, educational, and social services [32,34], but research has not examined how surveillance fears affect uptake of public health supports like home visiting.

Across three studies, we find that although home visiting is perceived as valuable, fear of being unfairly judged about parenting and the association of the term "home visit" with CPS impede participation among Black families. Through exploratory focus groups, Study 1 reveals trust-related barriers to engagement in home visiting centered around fears of surveillance. Study 2 quantifies the relation between surveillance fears and home visiting via a national survey of Black parents with young children. Study 3 uses a pre-registered field experiment to test whether cuing fears of surveillance and using different program labels ("home visit" versus "new baby wellness") affect interest in home visiting, demonstrating that one small (yet far from sufficient) step toward increasing Black families' participation is to change the name.

Collectively, these three studies show how the broader social context impacts Black parents' engagement in maternal and infant health interventions and demonstrate how program labels hold substantive and interpretive meaning. Within a context of widespread surveillance of family life, a label that cues "surveillance" versus "well-being" affects how people decide to participate in public health interventions.

## Methods

We conducted focus groups (Study 1) followed by an online survey (Study 2) and finally a field experiment (Study 3) to examine Black parents' views of home visiting and potential barriers to participation. All three studies were approved by the Duke Institutional Review Board for human subjects research (2022-01115). Written informed consent was obtained from all participants.

## Study 1: Focus groups with Black parents: perceptions of and perceived barriers to home visiting programs

### Methods

**Participants.** To understand Black parents' perceptions of home visiting programs, we conducted seven focus groups with 27 participants (6 men, 20 women, and 1 non-binary person) in a southeastern U.S. city where home visiting is offered universally to all parents who give birth in local hospitals. Participants were recruited via flyers posted in community centers and emails from community partners. All participants were parents or caregivers (hereafter, parents) of children three years old or younger. More than half the sample (n = 16) had at least a bachelor's degree, and annual household incomes ranged from less than $10,000 (n = 1) to between $70,000 and $79,999 (n = 1), with most (n = 20) reporting annual incomes between $30,000 and $59,999 (see demographic details in Table A in S1 Appendix).

**Procedure.** Focus groups took place on Zoom from December 2021 through May 2022, lasting 45–60 minutes. All focus groups were conducted in English by Black co-authors of this article and a Black community-based practitioner, to ensure moderator-participant race matching. Participants received $40 Amazon gift cards. Notably, despite home visiting being universally offered, only one participant had received visits from a home visiting program. Thus, facilitators provided descriptions of home visiting programs and elicited participants' understandings of the potential benefits or risks of participating (see protocol in S1 Appendix).

**Analysis.** Focus groups were recorded, transcribed, and coded using thematic analysis. Codes were identified based on prior literature (deductive approach) and new themes emerging from the transcripts (inductive approach). Fifteen percent of the transcripts were coded by multiple analysts, with disagreements resolved by consensus. Codes were then sorted to identity key themes.

**Results.** Participants' perceptions of home visiting were mixed. In response to facilitators' descriptions of home visiting programs, participants described home visiting as a helpful addition to the support already received from family members, and a more convenient, personally tailored, and COVID-friendly alternative to traveling to a medical center for support. However, parents expressed concerns over trust-related barriers to home visiting as focus group sessions progressed, describing hesitancy to receive home visits due to fears of being unfairly judged or concerns of reporting to CPS. One mother described such a worry: "they're going to think I'm doing something wrong to this baby…if [he] cries too long, they're going to come get me."

Trust-related barriers were classified into three categories:

1: Concerns About Poor Quality Treatment Due to Racism. Participants acknowledged a common understanding that Black individuals are subject to suboptimal treatment when accessing healthcare. One parent noted: "if a White and a Black person came to the hospital at the same time, the one who will be given attention first is the White person…if a White person [came] to my house, I don't think [they would] give me…[the quality] care [that I would have] if I was given a Black person to come to my home." Another alluding to similar concerns said, "[there is an] understanding that if you're Black you don't let White strangers into your home—bad things happen." Participants' concerns about poor quality treatment and racial discrimination in healthcare settings were not just hypothetical—many shared specific examples from their own experience. For instance, one parent described being left unattended in the hospital hallway for hours after reporting to the nearest emergency room for an ectopic pregnancy. Afterwards, that same participant reported preferring more distant hospitals for emergency care: "I don't mess with [the offending hospital that is closer to their home]…I want [the farther hospital]." When specifically asked whether they would take the provider's advice to participate in new baby wellness programs after experiencing provider mistreatment, participants also indicated that they would not seek out such programming. Instances like these shaped participants' subsequent interactions with healthcare systems, including decisions about whether to participate in programs related to maternal and infant health, such as newborn home visiting programs.

2: Explicit Fears of Surveillance. Participants discussed explicit fears of surveillance related to home visit participation, including (1) being regarded as a subpar parent, (2) being regarded as having unsatisfactory living conditions, and (3) being reported to government agencies such as CPS and Immigration & Customs Enforcement. Some participants believed that home visiting staff were tasked with in-home assessments concerning "the wellness of the family, the child, umm, err the cleaning, the cleaning." Another parent suggested that home visits carry negative connotations when the visit is initiated after someone "might have called" or filed a negative report about the child's well-being.

3: *Implicit Fears of Surveillance.* Concerns about surveillance also emerged implicitly when participants discussed needing more information before agreeing to participate. While not mentioning surveillance fears outright, these comments about perceived barriers nonetheless reflect *suspicions* of *s*urveillance. For instance, parents expressed concerns regarding lack of familiarity with home visiting staff and aversion to strangers visiting one's home. One participant discussed the uncertainty "of the intentions of that person [the home visitor]." Parents wanted to know about program procedures before deciding to participate and expressed a need to be given sufficient notice of the visit. While past research has found that scheduling challenges can pose a barrier to home visiting participation [19,35], participants here suggested that scheduling concerns were less about fitting visits into their schedules, and more about being granted sufficient notice "early [enough] before [the visit date] so that I can…be prepared for visits." Parents also implicitly acknowledged the power imbalance between home visitors and parents and desired "clear channels…to relay your complaints or for you to give feedback."

In sum, Study 1 illuminates how concerns about racially biased systems of care, coupled with surveillance fears, influence Black parents' views of home visiting services, even while many participants also acknowledged the potential benefits of home visiting. The role of discrimination experiences in shaping subsequent interactions with healthcare systems has been well documented and applies broadly to many healthcare services [36–38]. In contrast, surveillance fears are less well understood and uniquely relevant for home visiting programs, given the focus on the home context as the site of service delivery. Thus, we next sought to quantify the relation between surveillance fears specifically and home visit engagement within a larger, socioeconomically and geographically diverse sample of Black parents.

## Study 2: Survey of black parents' perceptions of home visiting

Study 2 examined the empirical relation between surveillance fears and participation in home visiting in a subsample of Black parents who participated in a nationwide online survey of parents.

### Methods

**Participants.** Participants were recruited in May 2022 by CloudResearch, an online research panel intended for scientific research. Participants in CloudResearch are prescreened at random times, not connected to any specific survey, which minimizes the incentive to falsify demographic data. Participation was limited to U.S.-residing, English-speaking parents or caregivers of child(ren) three years old or younger, including expectant parents. Of those recruited, 1,305 passed eligibility screening questions, and of those, 1,132 passed quality control checks described in S1 Appendix. We focus our analysis on the 163 Black respondents (76% women, 24% men, < 1% non-binary, residing in 35 states). As shown in Table B in S1 Appendix, the sample was socioeconomically diverse: 4% had not completed high school, 32% had a high school diploma or equivalent, 29% had attained some college education (but no degree), and 34% had an associate degree, bachelor's degree, or graduate degree. Annual household incomes ranged from $10,000 or less (10%) to $80,000 or more (12%).

**Procedure.** After eligibility screening, participants were asked to describe what comes to mind when they hear "home visits" (an open-ended question). Participants were then provided with a brief description of home visiting programs (see text in Section 2 of S1 Appendix) and asked about previous participation in such programs (yes/no). Finally, participants were asked what might prevent them from participating in home visiting via four items capturing trust-related barriers to engagement (e.g., "lack of trust," "fear of being judged") adapted from a widely used measure of obstacles to engagement in parenting programs [35]. Response options ranged from 1 (definitely not) to 4 (definitely yes), and the composite score was constructed by taking the mean across all five items ($M = 2.48$, Cronbach's alpha = 0.87).

**Analysis.** Responses to the open-ended question "what comes to mind…" were coded as representing surveillance fears if they indicated discomfort or fear (e.g., "intrusive," "in trouble"), judgment of parenting skills (e.g., "someone coming to determine if you are raising your child right"), or child removal (e.g., "taking children away"), or if they named CPS or other state-specific child welfare agencies. Coding was conducted independently by two research assistants with high inter-rater reliability (kappa = 0.70). Next, regression models analyzed the relation between associating home visits with surveillance and home visit engagement. In these models, the independent variable was a binary indicator equal to one if the participant associated "home visits" with surveillance or equal to zero if not, and the dependent variables were prior participation in home visiting programs and trust-related obstacles to engagement. Separate models were used for each dependent variable (details in Section 2 of S1 Appendix). All analyses were conducted in Stata 15.

**Results.** Twenty-five percent of parents associated the term "home visits" with surveillance. Moreover, participants who associated "home visits" with surveillance were nearly half as likely to report having ever participated in a home visiting program: 36% of parents who did *not* associate "home visits" with surveillance had participated in home visiting, compared with 17.5% of those who did (B = − 18.3 percent probability of receiving home visits, SE = 0.084, $p = .03$). Surveillance fears also predicted greater trust-related obstacles to engagement in home visiting (B = .40 SD, SE = 0.18, $p = .03$). Both associations were robust to the inclusion of individual-level factors previously linked to home visiting

enrollment: educational attainment [17], household income [39], help-seeking stigma [18], and housing instability [19] (Table D of S1 Appendix).

Study 2 provides evidence that surveillance fears are (1) common among Black parents from diverse socioeconomic and geographic contexts, and (2) linked to decreased participation in home visiting and greater trust-related barriers to engagement. We next sought to build on these correlational findings by testing whether surveillance fears cause reduced interest in home visiting among Black parents, and by examining which groups may be more (or less) affected by surveillance fears.

## Study 3: Experimental impacts of priming fears of surveillance and labeling of home visits on interest in learning more and participating in home visiting

Study 3 used a pre-registered field experiment to test the causal effect of a news story designed to prime surveillance fears on interest in home visiting and to examine the effect of changing the program label from "home visiting" to a "new baby wellness" visit using a between-subject design (pre-registration https://doi.org/10.17605/OSF.IO/TV2S6). We hypothesized that the surveillance prime would decrease interest, whereas language associated with supporting child wellness rather than emphasizing the home would make surveillance fears less salient, thus increasing interest.

Study 3 also included pre-registered analyses of moderation and mediation. We examined the moderating role of gender since the literature outlines persistent gender differences in child-rearing responsibilities [40,41], such that mothers are more likely to interact with mandated reporters at children's schools and medical appointments. In addition, existing research on hypervigilance and parenting surveillance focuses on mothers [32,34]. Taken together, these literatures suggest that women may be more affected by surveillance fears than men. We also examined moderation by individual-level factors associated with decreased home visit participation (low education, financial strain, and help-seeking stigma) [17,18,39], and by experiences of discrimination in healthcare settings based on Study 1 findings and literature showing how discrimination experiences shape interactions with healthcare systems [36–38]. Participants' prior experience with home visiting was also examined as a moderator since past program experiences (positive or negative) likely shape interest in subsequent programs. Finally, building on the role of trust in shaping perceptions of home visits observed in Study 1 and the association between surveillance fears and trust-related barriers to engagement observed in Study 2, mediation models examined trust-related barriers to engagement as a mechanism through which surveillance fears affect interest in home visiting.

### Methods

**Participants.** Participants were recruited by CloudResearch in May 2023. Participation was limited to U.S.-residing, Black or African American–identifying, English-speaking parents of child(ren) three years old or younger. Of those recruited, 1,513 passed pre-registered eligibility screening questions, and 1,094 passed pre-registered quality control checks (described in S1 Appendix). This sample size was based on a priori power calculations indicating a sample of 1,000 to detect to relatively small effects (f = 0.1) with alpha 0.04, and power 0.90, for main effects and interactions. Study 3 participants were 32 years old on average. Seventy-five percent were female, 24% were male, and less than 1% were non-binary. The sample was socioeconomically and educationally diverse; 26% had a high school diploma or less, 24% had at least a bachelor's degree, 42% had annual household incomes less than $40,000, and 14% were high-income (with annual household incomes of $80,000 or more) (see details in Table E in S1 Appendix).

**Procedure.** Participants were randomly assigned to read one of two newspaper articles. Those in the experimental condition read about a mother who experienced an unsubstantiated investigation by CPS, designed to cue surveillance fears. Those in the control condition read an article about excessive communication from schools (see article text in Section 3 of S1 Appendix). After reading, participants wrote a brief response describing how they would feel if they were the parent in the article, which served to enhance the personal salience of the article and was used as a manipulation check. Participants were then randomly assigned to read either a short description of "home visiting" or "new baby wellness" programs; all other language in the two descriptions was identical aside from the program label. Next, participants were asked about their interest in learning more and participating in the programs, were given the option of clicking a link to learn more (a behavioral measure of interest), and completed additional questions about basic demographics and healthcare experiences. Participants received compensation following CloudResearch protocols for their participation, which took approximately 15 minutes.

**Analysis.** Regression models measured the effect of the surveillance prime (versus child-centric news article) and the "new baby wellness" (versus "home visiting") label on interest in learning more and participating as well as the interaction between the two experimental conditions. Pre-registered covariates (age, household size, educational attainment, number of children, household income, immigrant status, and U.S. region of residence) were included for precision. Moderation was examined via interaction terms of the treatment indicators with pre-registered moderators, and the role of trust-related obstacles to engagement as a mediator was examined via a pre-registered structural equation model (details in S1 Appendix). All analyses were conducted in Stata 15.

**Results.** As hypothesized, the news story about unsubstantiated reports of child abuse decreased interest in learning more and participating in home visiting programs relative to the child-centered, neutral news story. Likewise, describing home visiting as a "new baby wellness" program rather than a "home visiting" program increased interest in learning more and participating (see regression output in Table F in S1 Appendix, and analyses of all pre-registered treatment contrasts in Table G in S1 Appendix). Interest in learning more and participating were both scored on a scale from 0 (not interested) to 100 (highly interested), with $M_{learningmore}$ = 62 and $M_{participating}$ = 69 among those assigned to the control conditions. The "new baby wellness" label increased interest in learning more and participating by 9–12 percentage points (two-sided and one-sided $p$s < .001), while the surveillance prime decreased interest in learning more and participating by 5 percentage points (one-sided $p$s, which are consistent with our pre-registered directional hypothesis, were .03; two-sided $p$s were .06). The magnitude of the effects of the "new baby wellness" label was significantly larger than the magnitude of the effect of the surveillance prime ($p$s < .01; see Table G in S1 Appendix). There was no evidence of an interaction between the two experimental conditions (Table F in S1 Appendix).

Pre-registered analyses of moderation revealed differential treatment effects by gender, educational attainment, and financial strain. The surveillance news story *decreased* the likelihood of clicking to learn more about the program among women, whereas it *increased* click-throughs for men (Tables H and I in S1 Appendix). The "new baby wellness" label increased interest in participating more for participants with high school or less educational attainment versus at least some college. In terms of financial strain, the "new baby wellness" label was most effective in increasing interest in learning more and participating among participants experiencing greater financial strain (i.e., worrying about meeting bills frequently versus sometimes or rarely). There was no evidence of moderation by our other pre-registered moderators: fear of judgment for help seeking, prior experience in home visiting, or experiences of discrimination in medical settings.

We also examined the role of trust-related obstacles to engagement in mediating the effects of both experimental conditions in a pre-registered structural equation model. This model tested the hypothesis that the news story designed to cue surveillance fears would *increase* trust-related obstacles to engagement, leading to *decreased* interest in home visiting, whereas the new baby wellness label would *decrease* trust-related obstacles to engagement, leading to *increased* interest in home visiting programs. Results revealed partial support for this hypothesis (Fig 1, and Table K in S1 Appendix). Trust-related obstacles to engagement mediated 11%–12% of the total effect of the "new baby wellness" label on interest in home visiting. However, there was no evidence that trust-related obstacles to engagement mediated the effect of the surveillance prime.

Structural equation model examining the direct and indirect pathways leading from the surveillance prime and the "new baby wellness" label to interest in learning more and participating in home visiting programs via trust-related obstacles to engagement. Estimated using maximum likelihood with robust standard errors. Standardized coefficients shown. Model includes covariance between learning more and participating (not shown).

Study 3 provides causal evidence linking surveillance fears to interest in home visiting among Black parents. Labeling home visits as a "new baby wellness" program increased interest in learning more and participating relative to the "home visiting" label. Likewise, cuing fears of surveillance by reading a news story about CPS had a negative effect on interest in home visiting, though this effect was less than half the size of the effect of the "new baby wellness" label. It may be that Black parents rely on sources other than news articles, such as social media and word of mouth, to judge the trustworthiness of programs like home visiting. Further, trust-related obstacles to engagement partially mediated the effect of the "new baby wellness" label, but did not mediate the effect of the CPS news story. This suggests that the news story may have been less effective at priming surveillance fears.

Moderation analyses show that the effects of the label change were strongest for parents with limited formal education and greater financial strain. Low education and financial strain are both risk factors for involvement in CPS [42,43], which may make these groups more

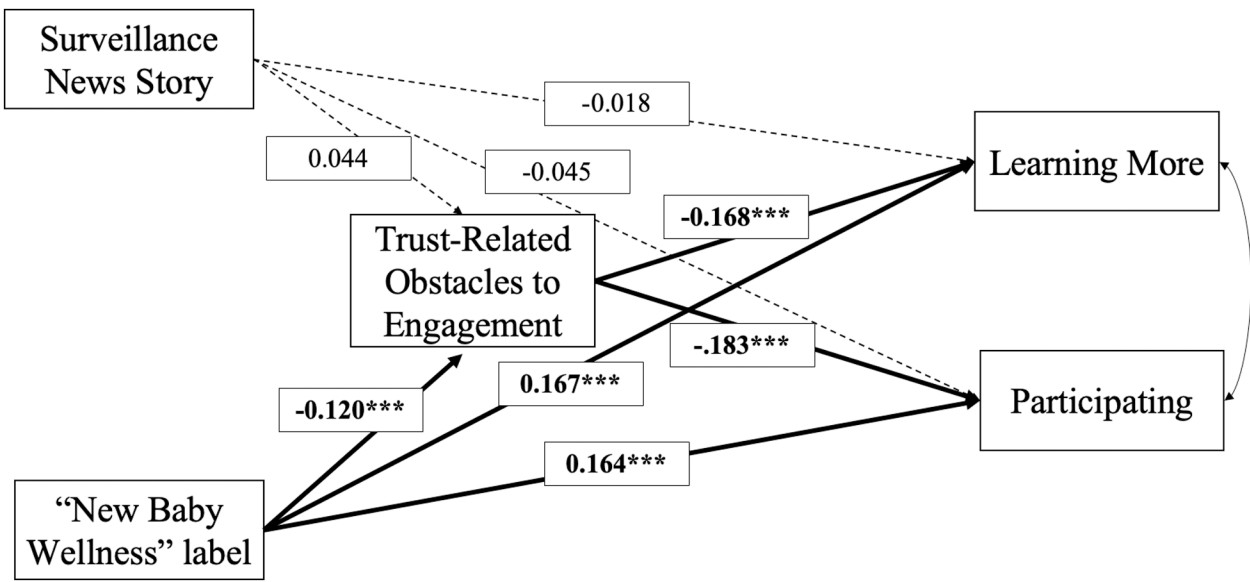

**Fig 1. Experimental effect of the "new baby wellness" (versus "home visiting") label on Black parents' interest in a newborn home visit program is partially mediated by a decrease in trust-related obstacles to engagement.**

aware of or concerned about surveillance relative to their peers with greater education and less financial strain. Moreover, even while home visiting is increasingly implemented universally, programs often prioritize families with limited socioeconomic resources who stand to benefit the most from connections to parenting resources [2]. Our findings suggest that it will be harder to reach these prioritized families with "business as usual" strategies that fail to address surveillance fears. Notably, there was no evidence of moderation according to experiences of discrimination in medical settings nor prior participation in home visiting, which highlights the universality of surveillance fears regardless of prior experiences with health services.

## General discussion

This paper finds that surveillance by the child welfare system has a chilling effect that reduces Black parents' engagement in programs intended to support child and family well-being during the critical "fourth trimester." Though Black focus group participants perceived positive qualities of newborn home visiting programs, they also expressed concerns about unfair judgment of their parenting skills (Study 1). Moreover, the term "home visits" elicited fears of surveillance via CPS. These themes were quantified in a larger sample of geographically and socioeconomically diverse Black parents (Study 2). One in four respondents associated home visiting with surveillance of family life, and parents who made this association were half as likely to have participated in home visiting programs compared with parents who did not. In a field experiment (Study 3), labeling a program a "new baby wellness" visit (rather than "home visiting") increased interest in home visiting, in part by reducing trust-related barriers to engagement. In contrast, reading a news story designed to cue surveillance fears (versus a neutral, child-centered news story) reduced Black parents' interest in home visiting, though this effect was smaller in magnitude than the positive effect of the "new baby wellness" label.

Collectively, our findings demonstrate the scientific value of centering the broader social context when designing and evaluating services intended to support maternal and infant health. We find that the context of surveillance from the child welfare system, itself embedded within a broader racialized climate, has negative effects on Black families' interest in home visiting programs. We do not find that this association is attenuated by more typically studied individual-level factors such as help-seeking stigma, educational attainment, or income.

Study 3 also shows the role of program labels in shaping how people understand and make decisions about enrolling in public health interventions. The label "new baby wellness visit," which centers the well-being of the newborn, increased parents' interest in learning more and participating relative to "home visiting," which centers the home and its condition. The effects of this name change were strongest for those deemed high priority for home visiting due to limited formal education and high financial strain [2]. However, we emphasize that more than a name change is needed to fully realize the promise of home visiting. Equitable service delivery requires a fuller appraisal of systemic bias and racism at all points of implementation, delivery, and interaction between families and programs.

### Limitations

This research was implemented in 2021–2022, a period of heighted concern about the COVID-19 pandemic and public discussion of racial injustices, contexts that may have impacted Black parents' interest in services delivered in their home and increased vigilance about protecting their families against racialized threats. It will be important to assess the temporal nature of findings in future work. In addition, this study's findings with a focus on Black families may not translate to families of other racial or ethnic groups or to parents with other marginalized identities.

## Public health implications

Home visiting programs receive substantial federal investments and have demonstrated potential to support healthy child development and family well-being [7]. Home visiting has been shown to reduce maternal and child mortality among Black women specifically [3], and indeed, the potential for home visiting to reduce racial health disparities is often cited as a reason to scale up evidence-based home visiting programs [10,11]. However, this potential promise of home visiting hinges on Black families' trust in such programs, which our findings suggest will require first addressing and mitigating concerns about surveillance.

## Supporting information

**S1 Appendix. Study procedure details and supplementary figures and tables.** (DOCX)

## Acknowledgments

The authors thank Mariette Aborn, Jackie Clay, Claire Morgan, and Brynne Townley for research assistance, Bernadette Greene for co-facilitating focus groups, and Karen Carmody, Caitlin Georgas, and Jessica McCoppin for their input on focus group protocol and survey measures. Community-based early childhood and maternal health organizations helped inform our research goals, and we are grateful for their support in recruiting Study 1 participants.

## Author contributions

**Conceptualization:** Jane Leer, Imari Z. Smith, Zoelene Hill, Lisa A. Gennetian.

**Data curation:** Jane Leer, Imari Z. Smith.

**Formal analysis:** Jane Leer, Imari Z. Smith, Zoelene Hill.

**Funding acquisition:** Jane Leer, Lisa A. Gennetian.

**Investigation:** Jane Leer, Imari Z. Smith, Zoelene Hill, Lisa A. Gennetian.

**Methodology:** Jane Leer, Imari Z. Smith, Zoelene Hill, Lisa A. Gennetian.

**Project administration:** Jane Leer.

**Supervision:** Lisa A. Gennetian.

**Writing – original draft:** Jane Leer, Imari Z. Smith, Zoelene Hill, Lisa A. Gennetian.

**Writing – review & editing:** Jane Leer, Imari Z. Smith, Zoelene Hill, Lisa A. Gennetian.

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
