## [Decision Letter · Decision Letter 0]

15 Oct 2024

PONE-D-24-41298Social contexts and Black families' engagement in early childhood programsPLOS ONE

Dear Dr. Leer,

Thank you for submitting your manuscript to PLOS ONE. After careful consideration, we feel that it has merit but does not fully meet PLOS ONE’s publication criteria as it currently stands. Therefore, we invite you to submit a revised version of the manuscript that addresses the points raised during the review process.

We look forward to receiving your revised manuscript.

Kind regards,

Julia Morgan

Academic Editor

PLOS ONE

Journal Requirements:

“The authors thank Mariette Aborn, Jackie Clay, Claire Morgan, and Brynne Townley for research assistance, Bernadette Greene for co-facilitating focus groups, and Karen Carmody, Caitlin Georgas, and Jessica McCoppin for their input on focus group protocol and survey measures. Community-based early childhood and maternal health organizations helped inform our research goals, and we are grateful for their support in recruiting Study 1 participants. This research was supported by a 2020 seed grant from All Children and Babies Thrive at Duke University and a 2022 State and Local Policy Work Grant from the Society for the Psychological Study of Societal Issues. “

 “This research was supported by an internal award from Duke University (2020 seed grant from All Children and Babies Thrive at Duke University) to Lisa A Gennetian: https://bassconnections.duke.edu/initiatives/abc-thrive. The funders did not play any role in study design, data collection and analysis, decision to publish, or preparation of the manuscript.”

5. We note that HV Paper_supp_PLOSONE in your submission contain copyrighted images. All PLOS content is published under the Creative Commons Attribution License (CC BY 4.0), which means that the manuscript, images, and Supporting Information files will be freely available online, and any third party is permitted to access, download, copy, distribute, and use these materials in any way, even commercially, with proper attribution. For more information, see our copyright guidelines: http://journals.plos.org/plosone/s/licenses-and-copyright.

a. You may seek permission from the original copyright holder of Figure(s) [#] to publish the content specifically under the CC BY 4.0 license.

Reviewers' comments:

Reviewer's Responses to Questions

**Comments to the Author**

1. Is the manuscript technically sound, and do the data support the conclusions?

Reviewer #1: Yes

Reviewer #2: Yes

2. Has the statistical analysis been performed appropriately and rigorously?

Reviewer #1: Yes

Reviewer #2: Yes

3. Have the authors made all data underlying the findings in their manuscript fully available?

Reviewer #1: Yes

Reviewer #2: Yes

4. Is the manuscript presented in an intelligible fashion and written in standard English?

Reviewer #1: Yes

Reviewer #2: Yes

5. Review Comments to the Author

Reviewer #1: Manuscript No: PONE-D-24-41298

Manuscript Title: Social contexts and Black families' engagement in early childhood programs

Study purpose: Given the potential positive benefit of home visiting programs, there is a need for deeper insights regarding the lower use/uptake by Black families. Through focus groups, survey data, and a pre-registered field experiment, the authors capture insights into the perception and experiences of Black parents with early childhood home visiting programs. The strength of this paper is numerous, especially through its use of diverse methods across the three studies. There are some areas the authors should address to strengthen its impact on the field.

The literature was concise and clear, attending to the current evidence and gaps in the evidence that this study sought to fill. The methods were described clearly, with some exceptions for some details being included in the narrative.

Concerning study 3, the authors make a note of moderators but don’t provide a compelling rationale or framework as to these moderators either in the introduction or the study section. Given there are likely moderators that generally make sense, it will be important for the authors to indicate why particular indicators are chosen and discuss them throughout the manuscript. For example, the moderators examined in Study 3 were not mentioned in the discussion. If available, these same moderators could have been valuable to look at studies 1 and 2.

One of the premises of this study is that parents’ concerns with surveillance were barriers to participating. This did not seem to be the case, at least in study 3. The authors should address the underlying premise, such as whether news stories may not be the tool for parents to interpret programs but potentially other sources, such as word of mouth, social media, etc. The authors must consider both the sociocultural context and the intersection with parents’ culturally grounded approaches.

Other thoughts:

p. 3, lines 64-65 – This sentence is a bit awkward. Are children investigated or families?

p. 5, lines 110-111 and p. 8, lines 188-189 – the authors can add a sentence or two about the demographics for the study 1 and 2 participants as they did with study 3 rather than readers going to the supplemental files.

Reviewer #2: I believe that the authors have touched on a very sensitive issue which is prevalent among some social contexts, in this case, Black families. The authors have conducted their research in the most sensitive and ethical manner and this has reflected in the final outcome. Participants were engaged and they were at ease with the research process. Findings from this research could have implications for practice, especially in re-naming of the researched programme to "New baby Wellness". This could make the programme more welcoming to the targeted groups and for them to enrol on the programme.

6. PLOS authors have the option to publish the peer review history of their article (what does this mean? ). If published, this will include your full peer review and any attached files.

**Do you want your identity to be public for this peer review?** For information about this choice, including consent withdrawal, please see our Privacy Policy .

Reviewer #1: **Yes: ** Iheoma U. Iruka

Reviewer #2: No

---

## [Author Response · Author response to Decision Letter 0]

10 Dec 2024

Response to Reviewers

Reviewer 1

Study purpose: Given the potential positive benefit of home visiting programs, there is a need for deeper insights regarding the lower use/uptake by Black families. Through focus groups, survey data, and a pre-registered field experiment, the authors capture insights into the perception and experiences of Black parents with early childhood home visiting programs. The strength of this paper is numerous, especially through its use of diverse methods across the three studies. There are some areas the authors should address to strengthen its impact on the field.

The literature was concise and clear, attending to the current evidence and gaps in the evidence that this study sought to fill. The methods were described clearly, with some exceptions for some details being included in the narrative.

Author response: Thank you for your positive feedback regarding the contribution of this manuscript. We are grateful for your attention to areas where we needed to add additional information (described in detail below).

Concerning study 3, the authors make a note of moderators but don’t provide a compelling rationale or framework as to these moderators either in the introduction or the study section. Given there are likely moderators that generally make sense, it will be important for the authors to indicate why particular indicators are chosen and discuss them throughout the manuscript. For example, the moderators examined in Study 3 were not mentioned in the discussion. If available, these same moderators could have been valuable to look at studies 1 and 2.

Author response: We added rationale for each of the pre-registered moderators in Study 3 and we have expanded the description of moderation findings in the discussion for Study 3 and the general discussion section.

In the introduction to Study 3 (page 14), we now say:

“Study 3 also included pre-registered analyses of moderation and mediation. We examined the moderating role of gender since the literature outlines persistent gender differences in child-rearing responsibilities [40,41], such that mothers are more likely to interact with mandated reporters at children’s schools and medical appointments. In addition, existing research on hypervigilance and parenting surveillance focuses on mothers [32,34]. Taken together, these literatures suggest that women may be more affected by surveillance fears than men. We also examined moderation by individual-level factors associated with decreased home visit participation (low education, financial strain, and help-seeking stigma) [17,18,39], and by experiences of discrimination in healthcare settings based on Study 1 findings and literature showing how discrimination experiences shape interactions with healthcare systems [36–38]. Participants’ prior experience with home visiting was also examined as a moderator since past program experiences (positive or negative) likely shape interest in subsequent programs.”

In the discussion for Study 3, page (19), we now say:

“Moderation analyses show that the effects of the label change were strongest for parents with limited formal education and greater financial strain. Low education and financial strain are both risk factors for involvement in CPS [42,43], which may make these groups more aware of or concerned about surveillance relative to their peers with greater education and less financial strain. Moreover, even while home visiting is increasingly implemented universally, programs often prioritize families with limited socioeconomic resources who stand to benefit the most from connections to parenting resources [2]. Our findings suggest that it will be harder to reach these prioritized families with “business as usual” strategies that fail to address surveillance fears. Notably, there was no evidence of moderation according to experiences of discrimination in medical settings nor prior participation in home visiting, which highlights the universality of surveillance fears regardless of prior experiences with health services.”

We return to these findings in the general discussion (page 20):

“The label “new baby wellness visit,” which centers the well-being of the newborn, increased parents’ interest in learning more and participating relative to “home visiting,” which centers the home and its condition. The effects of this name change were strongest for those deemed high priority for home visiting due to limited formal education and high financial strain [2].”

Lastly, we appreciate the suggestion to analyze the same moderators in Studies 1 and 2, but neither study was designed to assess moderation. Focus group participants were not selected with the aim of comparing between parents with different levels of education, income, help-seeking stigma, or discrimination experiences. Likewise, we are underpowered to assess moderation in Study 2. We believe it would be inappropriate to do so given these limitations. When describing Study 1 findings, however, we have expanded on experiences of discrimination, to make it clear why this moderator was pre-registered for Study 3. See pages 8–9:

“Participants’ concerns about poor quality treatment and racial discrimination in healthcare settings were not just hypothetical—many shared specific examples from their own experience. For instance, one parent described being left unattended in the hospital hallway for hours after reporting to the nearest emergency room for an ectopic pregnancy. Afterwards, that same participant reported preferring more distant hospitals for emergency care: “I don't mess with [the offending hospital that is closer to their home]….I want [the farther hospital].” When specifically asked whether they would take the provider’s advice to participate in new baby wellness programs after experiencing provider mistreatment, participants also indicated that they would not seek out such programming. Instances like these shaped participants’ subsequent interactions with healthcare systems, including decisions about whether to participate in programs related to maternal and infant health, such as newborn home visiting programs.”

One of the premises of this study is that parents’ concerns with surveillance were barriers to participating. This did not seem to be the case, at least in study 3. The authors should address the underlying premise, such as whether news stories may not be the tool for parents to interpret programs but potentially other sources, such as word of mouth, social media, etc. The authors must consider both the sociocultural context and the intersection with parents’ culturally grounded approaches.

Author response: Thank you for this helpful suggestion. We now expand on the limitations of using a news story to prime surveillance fears in the Discussion Section for Study 3, as follows (pages 18–19):

“…Likewise, cuing fears of surveillance by reading a news story about CPS had a negative effect on interest in home visiting, though this effect was less than half the size of the effect of the “new baby wellness” label. It may be that Black parents rely on sources other than news articles, such as social media and word of mouth, to judge the trustworthiness of programs like home visiting. Further, trust-related obstacles to engagement partially mediated the effect of the “new baby wellness” label, but did not mediate the effect of the CPS news story. This suggests that the news story may have been less effective at priming surveillance fears.”

Other thoughts:

p. 3, lines 64-65 – This sentence is a bit awkward. Are children investigated or families?

Author Response: We have revised this sentence (now page 4, lines 128–130) to read: “One in three children overall experience an investigation by Child Protective Services (CPS), compared with nearly half of Black children.”

p. 5, lines 110-111 and p. 8, lines 188-189 – the authors can add a sentence or two about the demographics for the study 1 and 2 participants as they did with study 3 rather than readers going to the supplemental files.

Author Response: We have added a sentence about the demographics for studies 1 and 2 to the main text.

Study 1: “More than half the sample (n = 16) had at least a bachelor’s degree, and annual household incomes ranged from less than $10,000 (n = 1) to between $70,000 and $79,999 (n = 1), with most (n = 20) reporting annual incomes between $30,000 and $59,999.”

Study 2: “We focus our analysis on the 163 Black respondents (76% women, 24% men, <1% non-binary, residing in 35 states). As shown in Table B in SI Appendix, the sample was socioeconomically diverse: 4% had not completed high school, 32% had a high school diploma or equivalent, 29% had attained some college education (but no degree), and 34% had an associate degree, bachelor’s degree, or graduate degree. Annual household incomes ranged from $10,000 or less (10%) to $80,000 or more (12%).”

---

## [Editor Report · Decision Letter 1]

16 Dec 2024

Social Contexts and Black Families' Engagement in Early Childhood Programs

PONE-D-24-41298R1

Dear Dr. Jane Leer, 

We’re pleased to inform you that your manuscript has been judged scientifically suitable for publication and will be formally accepted for publication once it meets all outstanding technical requirements.

Kind regards,

Julia Morgan

Academic Editor

PLOS ONE
---

## [Editor Report · Acceptance letter]

PONE-D-24-41298R1

PLOS ONE

Dear Dr. Leer,

I'm pleased to inform you that your manuscript has been deemed suitable for publication in PLOS ONE. Congratulations! Your manuscript is now being handed over to our production team.

Kind regards,

on behalf of

Dr. Julia Morgan

Academic Editor

PLOS ONE